# Optimal Duration of Antibiotic Therapy for Space Infections in the Maxillofacial Region: A Systematic Review

**DOI:** 10.3390/cmtr18030031

**Published:** 2025-07-03

**Authors:** Abdullah Saleh Alhudaithi, Faris Jaser Almutairi, Abdullah Saleh Almansour, Abdurrahman Abdurrazzaq Aljeadi, Shaul Hameed Kolarkodi

**Affiliations:** 1Medical City Dental Clinics, College of Dentistry, Qassim University, Qassim 51452, Saudi Arabia; alhudaithia.a@gmail.com; 2Department of Oral and Maxillofacial Surgery, College of Dentistry, Qassim University, Qassim 51452, Saudi Arabia; 3King Saud bin Abdulaziz University for Health Sciences, Riyadh 1461, Saudi Arabia; abdullahsalehalm@gmail.com; 4Ministry of National Guard, Riyadh 11426, Saudi Arabia; aaljeadi@gmail.com; 5Department of Oral and Maxillofacial Diagnostic Sciences, College of Dentistry, Qassim University, Qassim 51452, Saudi Arabia; s.kodi@qu.edu.sa

**Keywords:** antibiotic therapy, dentoalveolar, head and neck infections, maxillofacial space infections, odontogenic infections

## Abstract

Objective: This review aimed to examine and highlight the treatment protocols and optimal duration of antibiotic regimens used in managing maxillofacial space infections of odontogenic origin, along with the associated clinical outcomes. Materials and methods: This systematic review followed PRISMA guidelines and was registered in PROSPERO (CRD42024621000). A comprehensive search of PubMed, Scopus, Web of Science, and Google Scholar was conducted for studies from January 2003 to October 2024 using relevant MeSH terms. Studies were selected based on PEO criteria, focusing on the antibiotic treatment protocols and duration for odontogenic maxillofacial space infections, with inclusion of original human research and exclusion of non-relevant or unclear studies. Two independent reviewers performed study selection, data extraction, and risk of bias assessments using the Cochrane RoB 2 and ROBINS-I tools, resolving disagreements through discussion. Results: After data extraction, 277 papers were initially identified. Following the removal of duplicates, 141 articles were screened, of which 64 were selected for full-text assessment and 55 were excluded with justification. Ultimately, nine studies met the inclusion criteria for this review. These included two prospective double-blinded randomized clinical trials (RCTs), two prospective RCTs, four retrospective studies, and one prospective study, all involving patients with dentoalveolar orofacial infections. Risk of bias (RoB) assessment using RoB 2 indicated that two RCTs had a high risk of bias, one had a low risk, and one raised some concerns. ROBINS-I assessment showed moderate risk of bias in three studies, while two were not evaluated. Conclusion: This review concludes that prompt incision and drainage combined with a short-course antibiotic regimen of two to five days is generally effective for managing odontogenic maxillofacial space infections, though further high-quality randomized trials are needed to standardize treatment protocols.

## 1. Introduction

Orofacial infections are often classified as odontogenic or nonodontogenic. Infections arising from a tooth and its components are termed odontogenic infections (OIs). Conversely, dental structures are not implicated in nonodontogenic infections [1,2]. Dental caries, pulpal necrosis, dental trauma, and periodontal diseases can lead to OI, potentially resulting in serious repercussions on the soft and hard tissues of the oral cavity. A prior investigation indicated that Gram-negative bacilli were observed in 25% of patient oral specimens, while Gram-positive cocci played a key role in around 65% of OI [3].

Dental infections typically manifest as pain and edema in the oral cavity and must be addressed promptly, since they can result in severe and irreversible outcomes, including septicemia, osteomyelitis, airway blockage, cavernous sinus thrombosis, carotid infection, sinusitis, meningitis, orbital abscesses, and impaired vision [4]. The predominant characteristic of OI is the dentoalveolar abscess [3]. OI may be treated through surgical treatments, endodontic procedures, and prescription antibiotics. Prompt surgical intervention for the infected tooth is essential to avert additional complications. This may involve the removal of debris, irrigation, and drainage through an incision (I&D) in severe circumstances [5]. Additionally, in those exhibiting symptoms of systemic manifestations, the treatment of parenteral antibiotics based on bacteria culture results and sensitivity is recommended [4,6]. The present recommendations stipulate that antibiotics should be administered only after the eradication of infectious causes. These should be administered for two to three consecutive days following surgical interventions. Extended durations of antibiotic medication are not deemed sufficiently advantageous and are not advised [7]. They may lead to superfluous prescriptions and an extended course of antibiotic treatment, perhaps resulting in severe repercussions [8].

Antibiotic prescriptions may lead to unwanted effects, including hypersensitivity responses and dermatological or allergy problems [9]. Moreover, the unwarranted prescription of antibiotics may lead to several complications, such as bacterial resistance, gastrointestinal and hematological issues, and disruption of bacterial flora [10]. Moreover, this may result in oral bacterial resistance, which is regarded as an escalating problem in dentistry and medicine. Antibiotics should be administered with a narrow spectrum and restricted to acute illnesses to mitigate these issues. Furthermore, additional education and research must be undertaken to mitigate and diminish the issue of antibiotic resistance [11].

Despite a notable reduction in the frequency of odontogenic infections attributable to enhanced dental care and antibiotic effectiveness, these infections may pose potentially fatal hazards due to patient ignorance, oversight of the general practitioners, antibiotic therapeutic failure, compromised immune systems, concurrent medical conditions, or the lack of adequate medical services in developing nations [12,13]. These infections can spread swiftly within hours or days, resulting in potential consequences such as respiratory obstruction, sepsis, necrotizing fasciitis, descending mediastinitis, cavernous sinus thrombosis, and pericarditis, and may be fatal [12,14,15]. It is essential to implement a thorough therapeutic strategy for these infections; thus, early detection and appropriate treatment must be initiated immediately to prevent life-threatening consequences [14,15]. While recent reviews, such as Ribeiro et al. have provided insights into the optimal duration of antimicrobial therapy for odontogenic infections, their focus was primarily on infections affecting the jaws, evaluating different antibiotic regimens and their effectiveness in preventing complications [16]. However, there remains a critical need for a more focused analysis of maxillofacial space infections, specifically examining treatment duration in the context of odontogenic origins, clinical severity, and the role of adjunctive surgical interventions. While Ribeiro et al. focused on antimicrobial therapy duration in general odontogenic jaw infections, the present study uniquely targeted space infections of odontogenic origin, specifically involving fascial spaces requiring intraoral or extraoral drainage [16]. This focused approach allows more tailored recommendations for surgical management and antibiotic stewardship.

This systematic review builds upon the findings of previous studies by incorporating a broader range of study designs, including randomized controlled trials and retrospective analyses, to provide a more comprehensive understanding of the clinical outcomes associated with different antibiotic durations. Additionally, our review uniquely emphasizes the impact of short-course antibiotic therapy (2–5 days) following surgical intervention, aligning with emerging evidence on minimizing bacterial resistance and adverse effects. By narrowing the scope to maxillofacial space infections, we aim to refine treatment guidelines for clinicians managing these complex cases, ensuring a balance between efficacy and antibiotic stewardship. This review aimed to emphasize and examine the treatment protocols and optimum duration of antibiotic regimen employed in the treatment of maxillofacial space infections of odontogenic origin and the resulting clinical consequences.

## 2. Materials and Methods

### 2.1. Review Protocol

The process of reporting the present systematic review adhered to the Preferred Reporting Items for Systematic Reviews and Meta-Analyses (PRISMA) guidelines [17]. This study was registered in PROSPERO under protocol number CRD42024621000.

### 2.2. Research Question

The research question was “What are the treatment protocol and optimum duration for administering antibiotics for empirical treatment of maxillofacial space infections of odontogenic origin and associated clinical consequences?”

### 2.3. Information Sources and Search Strategies

The following databases were searched for relevant literature published between January 2003 and October 2024: PubMed, Scopus, Web of Science, and Google Scholar. For PubMed, Medical Subject Headings (MeSH) were used where applicable. The primary MeSH terms included: anti-bacterial agents, abscess, cellulitis, odontogenic tumors, periapical periodontitis, and head and neck neoplasms. To broaden the search across all databases, free-text keywords were also used in combination, such as “antibiotic therapy,” “empirical antibiotic treatment,” “antibiotic regimen duration,” “odontogenic infections,” “maxillofacial space infections,” “dentoalveolar abscess,” “pericoronitis,” and “periapical infections.” Boolean operators (AND/OR) were applied to combine the search terms. The search was limited to human studies published in English. After duplicate removal, two independent reviewers screened all titles, abstracts, and full texts for inclusion. Reference lists of the selected articles were manually checked for additional studies. Disagreements were resolved through discussion. A summary of the study selection process is presented in Figure 1.

A total of 274 records were identified through electronic databases, with 3 additional records identified through manual searching, resulting in 277 total records. After the removal of 107 duplicates, 167 records remained for title and abstract screening. Of these, 26 were excluded based on relevance. Subsequently, 141 full-text articles were sought for retrieval. However, 77 articles could not be accessed due to paywall restrictions, broken or outdated links, lack of institutional access, or unavailability through interlibrary services. The remaining 64 articles were assessed for eligibility, out of which 55 were excluded for reasons such as not focusing on the duration of antibiotic therapy, targeting irrelevant populations, or primarily addressing antibiotic sensitivity testing. Additionally, one manually identified study was excluded following full-text assessment. Ultimately, 9 studies met the inclusion criteria and were included in the review. The study selection process is illustrated in Figure 1.

### 2.4. Eligibility Criteria

The PEO (population, exposure, and outcome) framework was used to guide study selection.

Population: Individuals diagnosed with maxillofacial space infections of odontogenic origin.Exposure: Administration of oral or parenteral antibiotics as part of the treatment for odontogenic and maxillofacial space infections.Outcome: Clinical outcomes associated with antibiotic therapy, specifically the effectiveness and optimal duration of antibiotic regimens used as adjuvant treatment following surgical intervention.

We included original research articles such as randomized controlled trials (RCTs), comparative trials, and prospective or retrospective studies that evaluated antibiotic use in the treatment of odontogenic maxillofacial space infections. Studies were excluded if they did not clearly identify the odontogenic origin of infection, lacked specific information on antibiotic regimens or routes of administration, or focused solely on antibiotic sensitivity testing without reporting treatment outcomes. Case reports, animal studies, and narrative or systematic reviews were also excluded.

### 2.5. Study Selection and Data Extraction

Each study was evaluated separately by two investigators for study design, objectives, clinical interventions, antibiotic regimens (including dosage, route, and duration), and clinical outcomes. One of the primary outcomes of interest was clinical improvement (CI), defined as the resolution or marked reduction in infection-related symptoms such as pain, swelling, trismus, fever, or discharge within the reported follow-up period. Where available, the timing and criteria used to define CI were also recorded.

### 2.6. Quality Assessment

The risk of bias (RoB) for RCTs was evaluated as low, with certain concerns, or high, based on an assessment of research quality under the criteria established by the *Cochrane Handbook for Systematic Reviews of Interventions* [18]. The categories covered in RoB 2 include all types of bias recognized to affect the results of RCTs. These biases include bias due to the randomization process, planned intervention bias, missing-data bias, outcome measurement bias, and bias in the selection of reported results. The RoB was evaluated utilizing the Cochrane Collaboration’s ROBINS-I framework for non-randomized research, categorizing bias as low, moderate, serious, critical, or lacking information [19]. The instrument comprises seven categories: confounding bias, participant selection bias, intervention categorization bias, variation from intended intervention bias, missing-data bias, outcome assessment bias, and selective reporting bias. Inconsistencies were reconciled through discussion.

## 3. Results

### 3.1. Selection of Studies

Following data extraction, 277 articles were initially identified for potential inclusion in this review. After removing duplicates, 141 full-text articles were sought for retrieval. However, 77 of these could not be accessed due to issues such as paywall restrictions, broken or outdated links, lack of institutional access, or unavailability through interlibrary services. As a result, only 64 full-text articles were successfully retrieved and evaluated for eligibility in this review. Nine papers satisfied the eligibility criteria and were incorporated into the review and quality assessment [20,21,22,23,24,25,26,27,28] (Table 1). These studies varied in design, methodology, and antibiotic protocols. Below is a detailed analysis of each included study, highlighting their methodology, findings and limitations.

Banerjee et al. conducted a retrospective, multicentric study analyzing the efficacy of cephalexin–clavulanic acid, co-amoxiclav, and cefuroxime in odontogenic infections [20]. The study found that cephalexin–clavulanic acid resulted in faster clinical improvement compared to co-amoxiclav and cefuroxime, particularly in resolving pain, trismus, and fever. However, the retrospective nature of the study and variability in treatment adherence may have introduced bias, and the study did not provide data on long-term recurrence rates. Keswani et al. performed a five-year retrospective study evaluating maxillofacial space infections in a tertiary center [21]. The findings indicated that early detection and administration of IV antibiotics with I&D led to resolution within 72 h, with extraoral I&D cases requiring extended antibiotic courses. Despite these findings, the lack of randomization and standardized antibiotic regimens limits the generalizability of the study.

Kumari et al. conducted a prospective randomized controlled trial (RCT) comparing I&D alone versus I&D with antibiotics in odontogenic infections [22]. The study observed no significant difference in outcomes between the two groups, suggesting that antibiotics may be unnecessary in some cases when drainage is effective. Bali et al. carried out a double-blind RCT evaluating whether metronidazole is necessary after I&D in odontogenic infections [23]. The study concluded that there was no additional benefit of metronidazole when standard amoxicillin–clavulanic acid therapy was used.

Cachovan et al. performed a phase II RCT comparing moxifloxacin and clindamycin for odontogenic abscesses [24]. The study reported that moxifloxacin provided similar outcomes, but was better tolerated than clindamycin. However, the small sample and absence of long-term follow-up on recurrence were notable limitations. Ellison conducted a retrospective study analyzing a three-day antibiotic regimen for acute dentoalveolar abscesses with systemic involvement [25]. The findings suggested that a three-day course of amoxicillin, metronidazole, or clindamycin therapy was sufficient post-I&D. However, the study lacked a comparison to longer antibiotic courses or alternative treatment strategies.

Matijevic et al. undertook a prospective comparative study evaluating amoxicillin and cephalexin for acute odontogenic infections [26]. The study found that both antibiotics had comparable recovery times, though cephalexin exhibited slightly better efficacy. However, the study did not analyze bacterial resistance profiles, which could have provided additional insights. Kuriyama et al. performed a retrospective study assessing the impact of penicillin-resistant bacteria on odontogenic infection outcomes [27]. The study found that penicillin resistance did not significantly impact treatment outcomes when proper drainage was performed. However, the lack of prospective resistance tracking and potential selection bias were limitations of the study. Al-Belasy et al. conducted a prospective RCT evaluating azithromycin versus erythromycin for acute infraorbital space infections [28]. The study found that azithromycin reduced swelling and pain more effectively than erythromycin. However, the small sample and limited evaluation of long-term recurrence were notable limitations.

The findings from the included studies suggest that short antibiotic courses (2–3 days) following proper I&D appear effective, as demonstrated in the studies by Ellison and Kumari et al. [22,25]. Extraoral I&D cases, however, require longer antibiotic courses, as shown in a study by Keswani et al. [21]. Cephalexin–clavulanic acid showed superior symptom relief, as observed in a study conducted by Banerjee et al. A study by Bali et al. found no benefit from metronidazole as an adjunct, while Kuriyama et al. and Matijevic et al. demonstrated that antibiotic choice does not significantly impact clinical improvement when surgical management is appropriate [23,26,27].

Table 1 presents the main characteristics of the included studies, including design, sample details, intervention type, and antibiotic regimens, while Table 2 presents the main clinical outcomes reported in each study.

Table 3 below provides an overview of the study findings and limitations of the included studies, offering a comparative perspective on their contributions to this systematic review. The studies varied in their choice of incision and drainage (I&D) approaches, with some specifying intraoral or extraoral techniques. The severity of infection influenced the selection of I&D, impacting the duration and necessity of antibiotic therapy.

### 3.2. Study Characteristics

There were two prospective, double-blinded, randomized clinical trials [22,23], two prospective randomized trials [22,28], four retrospective studies [20,21,25,27], and one prospective study [25]. All patients presented with OI, encompassing periapical, periodontal, or pericoronal abscesses. The reviewed studies were primarily on individuals over 17 years of age, while two research incorporated pediatric patients [21,22]. All trials included participants who displayed acute odontogenic infections accompanied by systemic manifestations, including maxillofacial space edema, elevated fever, and lymphadenopathy. All trials included surgical or dental procedures to resolve the infection etiology through extraction or incision/drainage, with the administration of antibiotics.

### 3.3. Antibiotic Protocols Utilized

The antibiotics utilized in the review comprised amoxicillin, amoxicillin with clavulanic acid (CV), cephalexin–CV, cefalexin, cefuroxime, clindamycin, azithromycin, erythromycin, amikacin, metronidazole, moxifloxacin, and phenoxymethylpenicillin.

Across the nine included studies, a total of 12 different antibiotics were utilized, either as monotherapy or in fixed-dose combinations. The antibiotics were grouped as follows:
Beta-lactams (used in eight studies): amoxicillin (±clavulanic acid), cephalexin (±CV), cefuroxime, and phenoxymethylpenicillin.Nitroimidazoles: metronidazole (used in five studies).Macrolides: azithromycin and erythromycin (used in two studies).Lincosamides: clindamycin (used in three studies).Fluoroquinolones: moxifloxacin (used in one study).Aminoglycosides: amikacin (used in one study).

Amoxicillin–clavulanic acid was the most frequently used antibiotic, reported in six of nine studies. Metronidazole was the second most common, either as part of combination therapy or, in one case, as monotherapy. Moxifloxacin, azithromycin, and amikacin were each used in single studies.

Administration routes included:
Oral: used in seven studies.Parenteral (IV): reported in two studies (Keswani et al., Bali et al.).Mixed: some protocols shifted from IV to oral based on clinical improvement.

Treatment duration ranged from 2 to 7 days, with the most common duration being 3–5 days. A study by Matijevic et al. administered antibiotics until clinical improvement without a fixed duration [26]. Ellison et al. [25] reported that a 3-day antibiotic regimen was effective for managing dentoalveolar abscesses, challenging the traditional 5–7-day course. Figure 2 presents the distribution of antibiotics used across the nine included studies, grouped by class.

### 3.4. Clinical Outcomes of Antibiotics for OI

All included studies demonstrated clinical improvement (CI) with various oral or parenteral antibiotic regimens combined with surgical interventions. CI was reported within 2–3 days in five studies [21,23,25,27,28] and within 5–7 days in four others [20,22,24,26]. Kumari et al. [22] found no statistically significant difference in recovery between patients who underwent incision and drainage (I&D) alone and those who also received a broad-spectrum antibiotic combination (amoxicillin with clavulanic acid and metronidazole).

Matijevic et al. [26] observed comparable symptom resolution times in patients treated with amoxicillin (4.47 ± 0.62 days) and cephalexin (4.67 ± 0.65 days), whereas patients receiving surgical intervention without antibiotics showed a slightly longer recovery (6.17 ± 0.81 days; *p* < 0.05). Cephalexin–clavulanic acid demonstrated similar overall efficacy to cefuroxime and co-amoxiclav, with superior outcomes in reducing fever, erythema, gingival bleeding, and trismus [20].

Severe infections responded well to conventional regimens including amoxicillin–clavulanic acid, metronidazole, and aminoglycosides, particularly when combined with supportive measures such as intravenous therapy, hydration, extraoral drainage, drain placement, and management of systemic comorbidities [21]. Azithromycin showed advantages over erythromycin in reducing soft tissue edema, ensuring better patient compliance due to its simpler dosing, and demonstrating greater acid stability [28].

Clinical protocols typically involved reassessment within 48 h. Patients who showed no improvement underwent further wound evaluation, with modifications to the antibiotic regimen guided by clinical and laboratory data, and when available, culture and sensitivity results [23].

The approach to I&D (intraoral vs. extraoral) varied by infection severity and location. Keswani et al. [21] and Kumari et al. [22] noted that extraoral drainage, often used in deep fascial infections, was associated with longer antibiotic courses and hospitalization, while intraoral drainage in localized infections often sufficed with shorter regimens (2–3 days). Kumari et al. further observed that intraoral I&D alone was often sufficient, with no significant advantage observed when antibiotics were added postoperatively [22].

While clinical improvement and resolution timelines were consistently reported across the included studies, data on other critical outcomes—such as adverse events, recurrence rates, need for retreatment, length of hospital stay, and treatment costs—were either limited or entirely absent. Only a few studies briefly addressed adverse effects. For instance, Cachovan et al. reported a higher incidence of gastrointestinal side effects (nausea and diarrhea) in the clindamycin group compared to moxifloxacin, while other studies did not systematically assess or report such events [24]. Similarly, while Bali et al. described reassessment and modification of antibiotic therapy within 48–72 h based on clinical and laboratory parameters, detailed documentation of retreatment or recurrence rates was lacking. None of the studies evaluated cost-effectiveness or economic burden, despite its relevance in the context of antibiotic stewardship [23].

### 3.5. Assessment of the Quality of the Examined Studies

The RoB 2 showed that two of the examined RCTs had high RoB [24,28], one had low RoB [23], and one had bias concerns [22]. Figure 3 shows RoB findings from RCTs of the reviewed research. Figure 4 illustrates the RoB across the RCTs. The ROBINS-I methodology indicated that three studies [21,25,26] exhibited moderate RoB, whereas two investigations [20,27] showed low RoB (Figure 5). Figure 6 illustrates the RoB across the non-randomized studies.

## 4. Discussion

This systematic review found no substantial differences in overall clinical improvement (CI) among the various antibiotics used for odontogenic infections (OIs); however, multiple studies indicated that short-course antibiotic therapy (2 to 5 days) was effective when combined with appropriate surgical intervention [20,21,23,24,25,26,27,28]. In particular, tooth extraction or soft-tissue incision and drainage (I&D) led to faster symptom resolution compared to endodontic approaches [22,27].

Antibiotic selection in dental practice is largely informed by antimicrobial-susceptibility data, typically derived from pus samples and aligned with established prescribing guidelines [29]. Australian dental therapeutic recommendations, for instance, are influenced by susceptibility profiles from European countries, including Russia [30], Romania [31], and broader European datasets [32]. Kuriyama et al. [27] reported significant penicillin resistance in OI cases treated with phenoxymethylpenicillin or amoxicillin. Despite this, CI was observed within 2–3 days, and no major differences in outcomes were found between patients with penicillin-resistant and penicillin-sensitive infections. This led the authors to question the need for antibiotics when adequate drainage is achievable [27]. However, in cases such as diffuse cellulitis or deep-space infections, where drainage is delayed or not feasible, antibiotics play a more critical role. In hospitalized patients with severe infections, resistance was linked to worse clinical outcomes [33]. For moderate extraoral swelling, particularly when drainage is possible and systemic health is uncompromised, the benefit of broad-spectrum antibiotics should be reevaluated [34].

Pyogenic orofacial infections of odontogenic origin can range from localized periapical abscesses to extensive deep-space infections. If untreated, these infections may spread into adjacent fascial compartments (e.g., masseteric, sublingual, submandibular, buccal, canine, temporal, and parapharyngeal), leading to significant complications. Prompt diagnosis and timely intervention remain critical. While modern antibiotic regimens have reduced complication rates, the role of surgical judgment in managing purulent infections remains central [35].

Standard management protocols—elimination of the source, I&D, and adjunctive antibiotics—remain the foundation of care [36,37]. Matijevic et al. [26] showed that amoxicillin and cephalexin had comparable clinical outcomes, although cephalexin demonstrated superior bacterial susceptibility (89.2% vs. 76.6%). These findings emphasize the importance of correct antibiotic selection, dosing, and duration. Despite being the third-highest antibiotic prescribers in outpatient settings in the U.S. [38], dental practitioners are estimated to issue 30%–85% of prescriptions inappropriately [39,40,41].

Amid rising concerns about resistance and cost, judicious antibiotic use is increasingly prioritized [42]. Current guidelines from the American Dental Association and Centre for Evidence-Based Dentistry recommend short-course penicillin or amoxicillin for immunocompetent patients with localized infections and systemic signs. Therapy should be reevaluated within 3 days, and antibiotics can be stopped 24 h after complete symptom resolution [42,43].

The I&D approach significantly influences antibiotic duration. Intraoral I&D is favored for localized infections and usually requires a shorter antibiotic course. Extraoral I&D is indicated for severe infections affecting deep spaces like the submandibular, parapharyngeal, or masticator regions and often necessitates IV antibiotics for 5–7 days. Keswani et al. [21] and Kumari et al. [22] confirmed that intraoral I&D cases typically resolved within 2–3 days, while extraoral cases required prolonged therapy. Bali et al. [23] noted that metronidazole was not always necessary post-drainage, supporting narrower-spectrum antibiotic use in select cases. This stratification reinforces antibiotic stewardship and promotes evidence-based empirical prescribing.

Compared to the review by Ribeiro et al., which included eight randomized clinical trials analyzing antimicrobial duration post-odontogenic intervention, our study included both RCTs and observational studies with a focused inclusion of fascial space infections. We further analyzed the relationship between the surgical drainage approach (intraoral vs. extraoral) and the duration of antibiotic use, which was not explored in Ribeiro’s review. Among the nine studies we included, Banerjee et al. was not included in Ribeiro’s review [16,20]. This study provided important recent data on surgical drainage in submandibular and buccal space infections, offering new insights into optimizing therapy duration post-surgery (Table 4).

Based on the reviewed evidence, clinical recommendations include reserving antibiotics for patients with systemic involvement or deep-space infections. For mild to moderate infections managed with effective I&D, a 2- to 3-day course of amoxicillin or amoxicillin–clavulanic acid is often sufficient. Severe infections may require IV antibiotics for up to 7 days. Alternative agents like clindamycin or moxifloxacin may be used in penicillin-allergic or resistant cases, consistent with studies by Ellison, Kumari, Keswani, Kuriyama, and Cachovan [21,22,24,25,27].

Despite promising findings, further studies are needed to refine treatment protocols. Future research should focus on standardized reporting of I&D type, infection severity, and treatment duration, and evaluate outcomes such as recurrence, adverse events, hospital stay length, and treatment costs. Advanced microbial profiling techniques may also help identify resistance genes and inform targeted therapy [34]. Narrow-spectrum antibiotics should be prioritized in healthy individuals’ post-drainage, while broad-spectrum combinations should be reserved for complex cases. Ultimately, antibiotic stewardship must remain a guiding principle in the empirical treatment of odontogenic maxillofacial infections. The protocol by Cuevas-Gonzalez et al., although relevant in scope, was excluded from this review as it did not provide the clinical outcome data necessary for synthesis. It remains a potentially valuable future source once results are published.

## 5. Conclusions

This review examined the CI of antibiotics in localized maxillofacial space infections with identified odontogenic sources. The cornerstone of managing infections in the odontogenic spaces is prompt and vigorous incision and drainage, accompanied by the elimination of the underlying cause. An appropriate short-course antibiotic for a duration of two to five days is an essential complement. The prompt initiation of antimicrobials following diagnosis and before surgery can reduce the duration of infection and mitigate risks related to it. The precise application of antibiotics is essential for managing dental infections; therefore, thorough antimicrobial prescription guidelines must be developed for dental practitioners.

## Figures and Tables

**Figure 1 cmtr-18-00031-f001:**
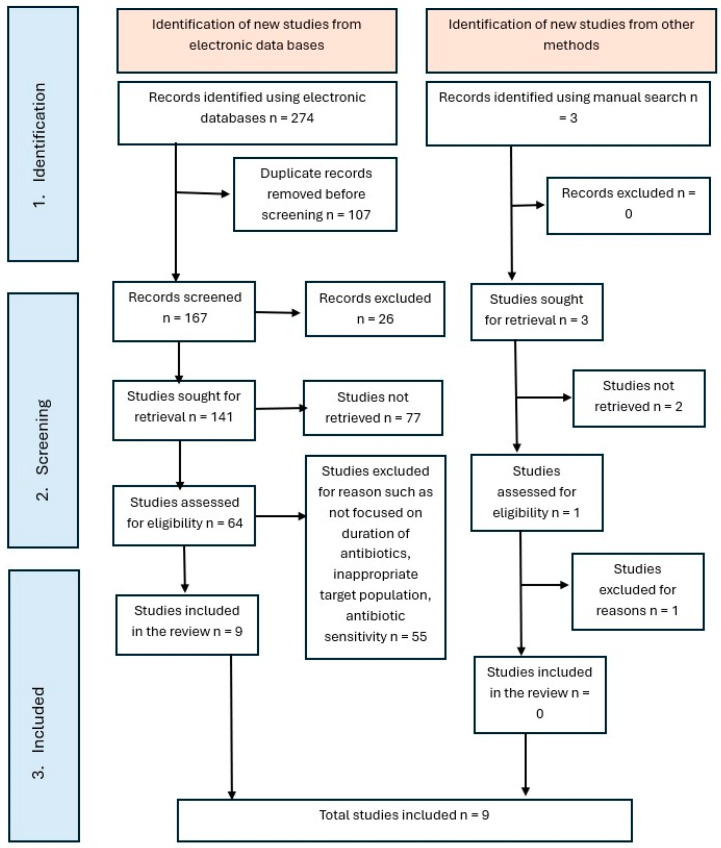
PRISMA flowchart of the reviewed studies.

**Figure 2 cmtr-18-00031-f002:**
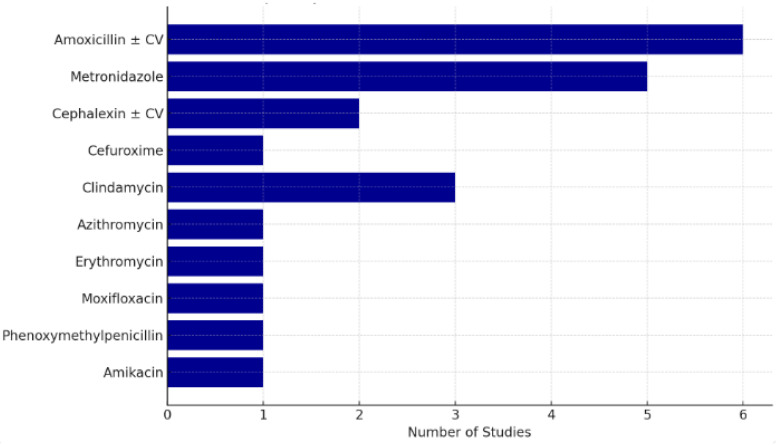
Frequency of antibiotics used across included studies.

**Figure 3 cmtr-18-00031-f003:**
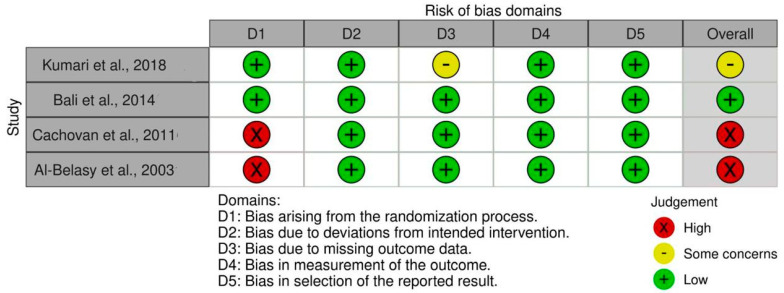
RoB 2 tool for evidence quality of the reviewed RCTs [22,23,24,28].

**Figure 4 cmtr-18-00031-f004:**
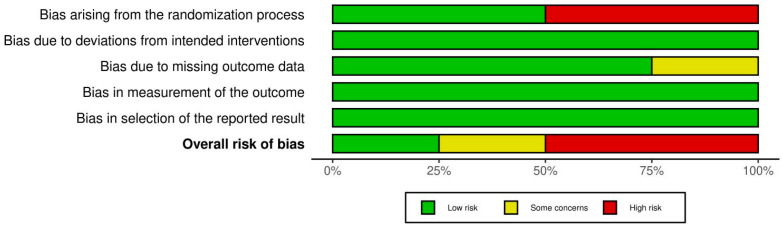
RoB 2 tool for evidence quality across the reviewed RCTs.

**Figure 5 cmtr-18-00031-f005:**
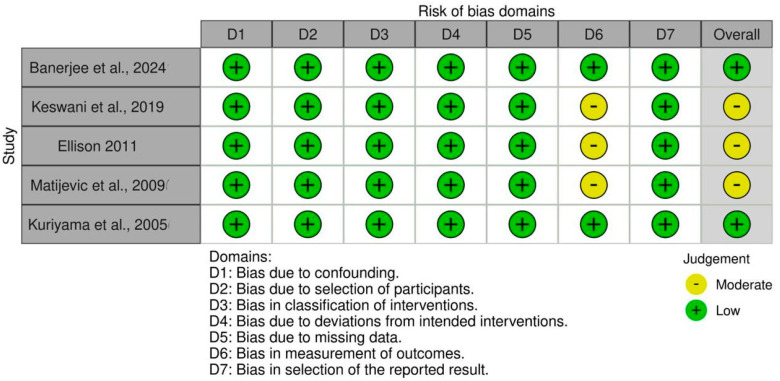
ROBINS-I tool for evidence quality of the reviewed non-randomized studies [20,21,25,26,27].

**Figure 6 cmtr-18-00031-f006:**
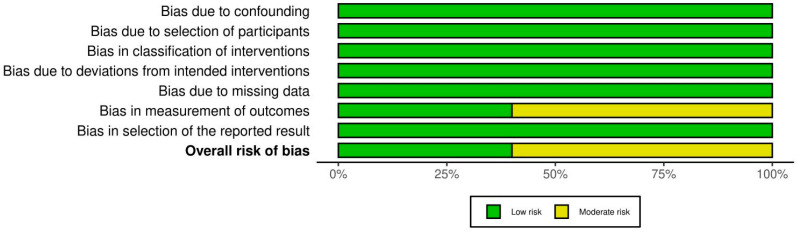
ROBINS-I tool for evidence quality across the reviewed non-randomized studies.

**Table 1 cmtr-18-00031-t001:** Characteristics of the reviewed studies.

Study	Study Design	Sample Size (Age Range/Mean)	Intervention Type	Antibiotics Used (Dosage, Route, Duration)
Banerjee et al., 2024 [20]	Retrospective, multicentric	355 adults (mean 39 years)	Oral antibiotics	Cephalexin–CV (375–750 + 125 mg); co-amoxiclav 625 mg; cefuroxime 250–500 mg for ~5 days
Keswani et al., 2019 [21]	Retrospective	315 (mean ~38 years)	Intraoral/extraoral I&D	IV amoxicillin–CV 1.2 g BD; metronidazole; amikacin
Kumari et al., 2018 [22]	RCT (Prospective)	40 (10–50 years, mean 27 years)	Extraction + I&D	Amoxicillin–CV 625 mg + metronidazole 400 mg TID vs. no antibiotics
Bali et al., 2014 [23]	RCT (double-blind)	60 (mean 33 years)	I&D	Amoxicillin–CV + metronidazole IV, 8-hourly
Cachovan et al., 2011 [24]	RCT (double-blind)	31 (>18 years)	Surgical + extraction	Clindamycin 300 mg QID or moxifloxacin 400 mg OD for 5 days
Ellison, 2011 [25]	Retrospective	188 (>18 years)	Drainage + extraction	Amoxicillin, metronidazole, clindamycin—all for 3 days
Matijevic et al., 2009 [26]	Prospective comparative	90	Extraction and/or I&D	Amoxicillin or cefalexin 500 mg QID for ~5 days
Kuriyama et al., 2005 [27]	Retrospective	112 (17–81 years)	Drainage	Multiple regimens—2–3 days
Al-Belasy et al., 2003 [28]	RCT (prospective)	60 (18–47 years)	Extraction ± I&D	Azithromycin 500 mg OD, erythromycin 250 mg QID, or none

CV—clavulanic acid; I&D—incision and drainage; IV—intravenous.

**Table 2 cmtr-18-00031-t002:** Study outcomes.

Author-Year	Outcome Summary
Banerjee et al. [20]	Cephalexin–CV showed faster symptom resolution than co-amoxiclav and cefuroxime.
Keswani et al. [21]	Infections resolved within 72 h with IV antibiotics and early intervention.
Kumari et al. [22]	Similar recovery in both groups; 75% showed drainage cessation within 3 days.
Bali et al. [23]	No difference in resolution between antibiotic combinations; reassessment done after 48–72 h.
Cachovan et al. [24]	Moxifloxacin had better tolerability than clindamycin; similar improvement in both groups.
Ellison [25]	Three-day antibiotics effective post-drainage for systemic dentoalveolar abscess.
Matijevic et al. [26]	Symptom duration ~4.5–4.7 days with antibiotics; surgery-only group took ~6.2 days to resolve.
Kuriyama et al. [27]	All regimens effective by 72 h; penicillin resistance did not affect outcomes.
Al-Belasy et al. [28]	Azithromycin showed better swelling reduction; both antibiotics improved outcomes by day 3.

**Table 3 cmtr-18-00031-t003:** Findings and limitations of included studies.

Study	Study Design	Sample Size	I&D Approach	Findings	Limitations
Banerjee et al. (2024) [20]	Retrospective	355 patients	Not specified	Cephalexin–clavulanic acid showed faster symptom resolution than co-amoxiclav.	Retrospective design; no randomization.
Keswani et al. (2019) [21]	Retrospective	315 patients	Extraoral	Extraoral I&D cases required longer antibiotic courses than intraoral cases.	No standardized protocol; lacks RCT design.
Kumari et al. (2018) [22]	RCT	40 patients	Intraoral	No significant difference between I&D alone and I&D with antibiotics.	Small sample; lacks subgroup analysis.
Bali et al. (2014) [23]	RCT	60 patients	Intraoral	Metronidazole offered no added benefit over amoxicillin-clavulanic acid alone.	No culture-based pathogen analysis.
Cachovan et al. (2011) [24]	RCT	31 patients	Not specified	Moxifloxacin and clindamycin had similar outcomes; moxifloxacin better tolerated.	Small sample; no long-term follow-up.
Ellison (2011) [25]	Retrospective	188 patients	Intraoral	Three-day antibiotic regimens were sufficient post-I&D for acute abscesses.	No comparison to longer regimens or alternatives.
Matijevic et al. (2009) [26]	Prospective comparative	90 patients	Not specified	Cephalexin and amoxicillin had similar outcomes; cephalexin slightly superior.	No bacterial resistance analysis.
Kuriyama et al. (2005) [27]	Retrospective	112 patients	Not specified	Penicillin resistance did not impact outcomes when drainage was performed.	No prospective tracking of resistance.
Al-Belasy et al. (2003) [28]	RCT	60 patients	Extraoral	Azithromycin more effective than erythromycin in reducing pain and swelling.	Small sample; limited long-term outcome data.

**Table 4 cmtr-18-00031-t004:** Comparative analysis of studies in Ribeiro et al. [16] and the current systematic review.

Study ID	Included in Ribeiro et al. [16]	Included in Current Review	Reason for Inclusion/Exclusion
Natarajan et al., [44]	Yes	Yes	Overlapping study on transalveolar extraction
Bali et al., [23]	Yes	Yes	Metronidazole-based therapy after I&D
Matijević et al., [26]	Yes	Yes	Acute dentoalveolar abscesses
Mohanty et al., [45]	Yes	Yes	Maxillofacial fractures and antibiotic course
Luaces-Rey et al., [46]	Yes	No	Focused on prophylactic regimens, not space infections
Arteagoitia et al., [47]	Yes	No	No data on fascial space involvement
Salim et al., [48]	Yes	No	Observational study with limited data on space infections
Banerjee et al., [20]	No	Yes	New study on submandibular and buccal infections

## Data Availability

No new data were created or analyzed in this study. Data sharing is not applicable to this article.

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
