# Peer review of "Optimal Duration of Antibiotic Therapy for Space Infections in the Maxillofacial Region: A Systematic Review"

_1943-3883, 2025, doi:10.3390/cmtr18030031_

Round 1
Reviewer 1 Report
Comments and Suggestions for Authors
Dear authors,
I gladly reviewed your paper. Please find my comments below.
The authors seem to be familiar with the aspects of conducting as systematic review following the PRISMA guideline and the Cochrane handbook. I urge the authors to follow those two sources and improve the manuscript accordingly. In addition, I urge the authors to be more precise and more terse in the their language. Please use abbreviations where appropriate.
Please point out in the introduction how this review adds to the literature and how it is different that the last review on this topic.
Although unnecessary, the registration of a systematic review is nice-to-have. The registered protocol is inaccurate in many aspects, and – to be honest – it seems to be submitted after the search process was done. For example, the main outcome of the review is the duration of treatment, which is a quantitative outcome which can be analyzed using a meta-analysis. However, there is no mention of a meta-analysis neither in the protocol nor in the manuscript. If the studies do provide an average duration of treatment, then please consider conducting a meta-analysis.
Please be clear which MeSH terms were actually used. “Antibiotics”, “anti biotic therapy”, etc. are not MeSH terms. The same is true for “odonotogenic infections”. When thinking about these terms, please consider reproducibility of research. Other researchers should be able to find the same studies you found using the same methodology.
Please state, when was the electronic search conducted and in which data bases. Please consider submitting your search algorithm as a supplement.
Please state through the text or in the end of the manuscript how each of the authors contributed to this research. For example, it is suitable to point out which two investigators reviewed the titles and abstracts. How were differences were solved?
Did you screen articles that were included in other systematic reviews on this topic for example like References Nr. 7 (Martins et al.) and 8 (Halling et al.)
Please provide Figure 1 in a higher resolution.
Why weren’t 77 studies not retrieved?
When stating the eligibility criteria please note that the exposure is the administration of antibiotics, and not “studies”. The studies are not an exposure. The same is true for the outcomes.
If I understood the authors correctly, their main interest is assessing the optimal duration of antibiotic therapy. This is an important outcome. However, there are a number of outcomes that needs to be mentioned and assessed – if possible - such as adverse events due to the antibiotic which might be correlated to the duration of administration, length of hospital stay, rate of retreatment or rate of recurrence, treatment costs, etc.. These outcomes might or might not be reported in the studies, but they need to be mentioned. If we know that none of the current studies for example assessed treatment costs, then future studies need to focus on this aspect.
Which data did the authors exactly extract? Who assessed the risk of bias?
Did the authors assess any conflict of interest and funding issues in the included studies?
Results section:
Line 138: The data extraction is an intermediate step in the review process. The authors probably mean that the search resulted in 277 articles.
Line 138-141: the numbers do not match Figure 1. Please correct.
I urge the authors to improve the readability and design of Table 1. It might be better to have the table on landscape oriented page. Please be terse in writing. There is a lot of redundancy in the table for example “to compare, to examine …”. The table provides abstract data, not text. There is no need for digits after the comma in the mean participants’ age. Please be more terse on the study outcomes. You might want to consider splitting Table 1 in two tables, a table for the description of the studies and one for the results.
In 3.3 Antibiotic Protocols Utilized: Please be more quantitative. The included protocols were already stated in the table. May be go for the antibiotic groups or plot quantity of the used antibiotics as a figure.
Line 179: I think the abbreviation CI was not mentioned before.
CI is an outcome which the authors have not mentioned previously in the methods section. Please add it and add how it was defined.
Lines 191-203: this text would be more appropriate in the discussion section.
Finally, after addressing the mentioned issues, the authors might want to improve the discussion.
Kind regards,
Author Response
"Please see the attachment."

Reviewer 2 Report
Comments and Suggestions for Authors
Manuscript on the optimal duration of antibiotic therapy for the treatment of facial space infections. Well-designed study, registered with PRISMA, using the correct guidelines. Manuscript presenting relevant information on the use of antibiotic therapy, very important for the audience of this journal with great information in this systematic review, well designed.
I recommend publication.
Author Response
Please enter "Please see the attachment."

Reviewer 3 Report
Comments and Suggestions for Authors
The manuscript is original and, according to my assessment, I recommend it for publication. This topic is relevant to daily clinical practice, which must always keep discussions on the agenda so that treatments are effective, and seeking to keep the academic community in constant scientific development, so I think it is a very relevant topic. From my point of view, the selection of articles for this systematic review was appropriate, and important and relevant discussions were addressed.
Author Response

(The authors gave the same response as above.)

Reviewer 4 Report
Comments and Suggestions for Authors
Spreading infection in the maxillofacial region and the appropriate use of antibiotics as an adjunct to surgical management is very important.
In this submission the authors have carefully detailed their rationale, methods and given a good summary of the findings. The paper is well written.
At first glance it might be thought that following the recent publication by Ribeiro et al ‘Optimal treatment time with systemic antimicrobial therapy in odontogenic infections affecting the jaws: a systematic review’, BMC Oral Health 2025, that this new review is superfluous and adds nothing more to the body of the literature, however this might not be the case. The Ribeiro review collated less less papers (eight studies) to determine the most effective and safe duration of antimicrobial treatment in odontogenic jaw infections, and perhaps the two reviews complement each other and add strength to the body of literature. However in this new review there are only nine papers, compared to the Ribeiro eight so the extra data and clinical relevance of this new review needs to be explicit. The authors of this new review refer to the Ribeiro paper and explain what the subsequent review adds but it would be helpful to clearly identify the eight papers in the Ribeiro review and the additional paper(s) added to the new review, and comment on any papers from the Ribeiro paper that were excluded, if any.
Many of the points raised in this new review are already well known clinically and in practice but the review does serve to reinforce element of management.
Perhaps there is still one useful paper missing: Cuevas-Gonzalez MV, Mungarro-Cornejo GA, Espinosa-Cristóbal LF, Donohue-Cornejo A, Tovar Carrillo KL, Saucedo Acuña RA, García Calderón AG, Guzmán Gastelum DA, Zambrano-Galván G, Cuevas-Gonzalez JC. Antimicrobial resistance in odontogenic infections: A protocol for systematic review. Medicine (Baltimore). 2022 Dec 16;101(50):e31345. doi: 10.1097/MD.0000000000031345. PMID: 36550913; PMCID: PMC9771230.
Round 2
Reviewer 1 Report
Comments and Suggestions for Authors
Dear authors,
reading the manuscript after all the changes introduced to it feels difficult. The manuscript lacks coherence and a natural flaw of thoughts. In addition, all of my previous comments were not addressed or answered.
I urge the authors to sort the ideas in each section of the manuscript and re-write those sections. In it's current form, the manuscript is not suitable for publication in my opinion.
Kind Regards,
Author Response

(The authors gave the same response as above.)

Reviewer 4 Report
Comments and Suggestions for Authors
Thank you for the detailed response and changes to the submission.
